# Long-Term Insomnia Treatment with Benzodiazepines and Alzheimer’s Disease: A Systematic Review

**DOI:** 10.3390/neurosci6010011

**Published:** 2025-02-01

**Authors:** Filipa Sofia Trigo, Nuno Cardoso Pinto, Maria Vaz Pato

**Affiliations:** 1Faculty of Health Sciences, University of Beira Interior, 6200-506 Covilhã, Portugal; a37438@fcsaude.ubi.pt (F.S.T.); nuno.pinto@fcsaude.ubi.pt (N.C.P.); 2CICS-UBI—Health Sciences Research Centre, University of Beira Interior, 6200-506 Covilhã, Portugal; 3Rise-Health, University of Beira Interior, 6200-506 Covilhã, Portugal

**Keywords:** Alzheimer, benzodiazepines, insomnia, systematic review

## Abstract

Alzheimer’s disease is the most common form of dementia. Benzodiazepines are the most widely used pharmacological class in the treatment of insomnia and other sleep disorders. Some literature suggests that the chronic use of benzodiazepines is associated with the development of cognitive decline. This review aims to evaluate the use of benzodiazepines and its association with the development of Alzheimer’s disease. A systematic review of the literature was carried out using the MEDLINE and Embase databases. Protocols followed the PRISMA-P 2020 methodology, and, after the analysis of the included studies, a narrative synthesis of the results was carried out. Only two cohort studies were identified that met defined eligibility criteria. In the retrospective study, a significant risk of developing Alzheimer’s disease after treatment with benzodiazepines was found. In the prospective study, the prevalence of Alzheimer’s disease was not associated with treatment with benzodiazepines. Results suggest that only the largest study presented a significant risk of developing Alzheimer’s disease. Given the scarce scientific evidence found, it is concluded that further research on this topic is necessary.

## 1. Introduction

Alzheimer’s dementia (AD) is the most common form of dementia, affecting over 55 million people worldwide [1,2]. It is a neurodegenerative disease characterized by insidious and progressive cognitive decline and is distinguished from other dementias by the unique cardinal signs of its histopathology [3]. Histopathological findings consist of the presence of extracellular neuritic plaques consisting of β-amyloid protein and intraneuronal neurofibrillary braids consisting of hyperphosphorylated tau protein strands associated with reactive dystrophic neuritis, astrogliosis, and microgliosis, culminating in synaptic and neuronal loss with consequent medial temporal atrophy and diffuse cortical atrophy [3,4,5].

The etiology of AD is associated with several risk factors. Age is the most important risk factor, with diagnosis being made mostly after the age of 65—late-onset AD. In turn, early-onset AD is a rare entity, mostly related to familial AD, of autosomal dominant transmission [6]. As for acquired or modifiable risk factors, studies highlight a low level of education, cerebrocardiovascular diseases, arterial hypertension, dyslipidemia, obesity, diabetes mellitus (DM), infections (particularly in the central nervous system (CNS)), smoking, and environmental pollution, as well as insomnia [6,7].

Insomnia is defined as a disturbance in the normal sleep pattern, characterized by difficulty in the initiation, consolidation, or quality of sleep associated with limitation and/or impairment in daytime activities. Consequently, decreased cognitive performance, fatigue, or mood disturbances may occur [8,9]. The worldwide prevalence may reach up to 20% of the population, depending on defined diagnostic criteria [10]. Considering the increasing trend in the prevalence of insomnia, there has consequently been a significant increase in the use of benzodiazepines (BZDs) and/or hypnotics [11].

Short-half-life BZDs are recommended in the treatment of insomnia due to their sleep-inducing property. The main effects are decreased sleep latency and increased total sleep duration [12]. The European guidelines for the diagnosis and treatment of insomnia recommend the use of BZD only when insomnia is pathological, with a maximum duration of 4 weeks, including the discontinuation period [11,13], emphasizing that the efficacy of BZD is only verified in short-term treatment (≤4 weeks). Short-half-life drugs should be preferred because they are less likely to cause sedation in the morning and are less prone to the development of withdrawal syndrome [11,12]. The main adverse effects resulting from the use of BZD are manifested by decreased vigilance, with consequent drowsiness, mental confusion, anterograde amnesia, and motor incoordination, which leads to increased risk of falls [11,14].

Regarding the elderly population, numerous adverse effects have been reported, with special emphasis on cognitive decline, amnesia, confusional state, abuse and dependence, delirium, breathing difficulties, ataxia, vertigo, falls and paradoxical reactions (sleep disturbance, anxiety, and agitation), and its use has been found to be related to an increased risk of suicide [15]. Crowe et al. described the negative long-term effects of BZD use in several domains, namely working memory, processing speed, divided attention, visual construction, short-term memory, and expressive language [16]. Paterniti et al. (2002) identified a positive association between cognitive changes in the elderly population and long-term BZD use, reporting a decrease in cognitive performance. This association was independent of other factors, including gender and educational level [17]. Billioti de Gage et al. (2012) found in a prospective study an association between BZD use and the development of dementia, describing that BZD use was associated with a 50% increased risk of developing dementia [18].

It is important to highlight that the association of BZD use with the development of dementia and AD is controversial. The potential link between long-term BZD use and neurodegeneration has been suggested, based on findings such as the apparent impairment of synaptic integrity (loss of neural connections) [19] or even the reduction in the hippocampal and amygdala volumes [20]. Given that insomnia and anxiety may present as prodromal symptoms of dementia and that BZDs are used in these conditions, the symptoms described could be associated with the early stages of dementia, so it is difficult to establish a causal relationship between the use of BZD and the development of dementia [13]. Regarding AD itself, there is little consensus as to whether there is a causal relationship between these two variables, with authors describing positive [21] and negative associations [22].

As the existing literature does not present a definitive association between AD and the use of BZD and considering the high incidence and prevalence of AD and its lack of curative treatment, it is important to attempt to identify if there is an association between the long-term use of BZD in the treatment of insomnia and AD. Therefore, the primary objective of this study is to evaluate the association of long-term treatment of insomnia with BZD and AD. Secondarily, the aim is to assess the characteristics of the sample that underwent treatment with BZD.

## 2. Materials and Methods

The present systematic review followed the PRISMA-P 2020 methodology [23]. Participants included in this review were adults (age 18 years and older) diagnosed with insomnia. The intervention consisted of the long-term administration of BZD (greater than 1 month), with adults diagnosed with insomnia in whom BZD use was zero or less than 1 month as the comparison. The primary outcome assessed in this review was the development of AD. The Prospero registration ID is 641385. Relevant information associated with the determination of the characteristics of the sample who underwent treatment with BZD, namely age, sex, housing conditions, level of education, socioeconomic status, comorbidities, and concomitant medication, were also analyzed. The eligibility criteria defined for this review are detailed below.

### 2.1. Inclusion Criteria

Adult participants (age 18 years or older) with a diagnosis of insomnia; long-term treatment with BZD (greater than 1 month); development of AD after treatment with BZD; prospective cohort studies and retrospective cohort studies; and the control groups accepted were participants who did not take long-term treatment with BZD.

### 2.2. Exclusion Criteria

Diagnosis of another type of dementia; diagnosis of another type of sleep disorder; absence of characterization of AD, specifically the absence of a diagnosis of AD or the diagnostic criteria for AD; absence of characterization of insomnia, specifically the absence of a diagnosis of insomnia or the diagnostic criteria for insomnia; diagnosis of AD prior to treatment with BZD; meta-analyses and systematic reviews of the literature; and studies not published in Portuguese, Spanish, English, French, or Italian.

### 2.3. Search

The search was based on the PubMed/MEDLINE and Embase databases in order to gather studies on insomnia, BZD use, and AD diagnosis. The PubMed/MEDLINE and Embase searches were performed until August 2021. A systematic review of the literature was carried out until August 2021 using the MEDLINE and Embase databases, developing specific combinations of keywords and expressions for each database. Trade names of BZDs were not searched given the disparity of nomenclature in different countries. The search strategy is detailed in Appendix A. A complementary search was also carried out based on other sources and bibliographic references of articles considered relevant, so as not to lose information.

### 2.4. Study Selection

After the main searches, duplicate articles were identified using Mendeley Desktop version 1.19.8. Then, all titles and abstracts of the article selection were systematically and independently evaluated by two reviewers (F.T. and M.V.P.). A third reviewer was called to resolve differences. After the title and abstract screening, the articles considered relevant were subsequently read in full, following the eligibility criteria. The authors of articles whose relevant information was not fully described were contacted, up to 26 November 2021, trying to obtain the needed information. Based on the data obtained, a new analysis of the articles was performed, selecting the final articles to be included in the systematic review.

### 2.5. Data Extraction

Data extraction was performed using a table for each of the studies included in the review, which shows the following: author, title, database, study type, country, language, publication date, journal of publication, study duration (follow-up), sample characterization, including the number of participants and controls, characterization of BZD treatment (sex and age; short-, intermediate-, and long-half-life BZD; and treatment duration); characterization of insomnia (initial and terminal insomnia; diagnostic criteria), characterization of AD (diagnostic criteria), and outcomes.

The assessment of the quality of the studies under analysis was assessed using the assessment tool from the Joanna Briggs Institute (JBI)—“JBI Critical Appraisal Checklist for cohort studies” [24].

The risk of bias of the studies under analysis was assessed using the Cochrane’s “Risk Of Bias In Non-Randomized Studies—of Interventions (ROBINS-I)” [25]. The figures corresponding to the risk of bias were created using the robvis (Risk Of Bias VISualization) tool [26].

### 2.6. Strategy for Data Synthesis

The strategy used was based on data from studies that were considered relevant, in a synthesized and concise manner. We defined a strategy for presenting the results, trying to group the synthesis according to two main factors: (i) the type of study and the possible use of control groups (and what type of control), given that their characteristics may limit the valuation and generalization of results; (ii) the way of presenting the main result (risk of developing AD) to associate parameters considered similar. The assessment was carried out quantitatively, with the risk of developing AD assessed according to the Hazard Ratio (HR) and respective confidence interval (95%), as well as the Odds Ratio (OR), depending on the available data. The HR and OR are important statistical measures that evaluate the relationship between variables and outcomes. Both ratios compare event likelihoods, but HRs show how quickly events occur over time, while ORs give a static association without considering time [27].

## 3. Results

### 3.1. Selection of Studies

From the search in the PubMed/MEDLINE and Embase databases, 840 results were obtained, with the results concerning each of the databases being 83 and 757 articles, respectively. After removing 72 duplicate articles, a total of 768 articles were obtained.

From the research carried out through bibliographical references in relevant articles on this subject, seven articles were obtained. After reading the title and the abstract of all the articles (768 + 7), a total of 16 articles were obtained, of which 1 was a poster and 2 were restricted from full reading. Afterwards, the selected articles were read in full, and the absence of detailed information regarding various criteria was noted when verifying the inclusion criteria.

Of the 16 authors contacted, a response was obtained for 4 of the 16 articles. The author whose article corresponded to a poster was contacted, to verify the existence of a corresponding study. After reading the articles with the data obtained after contact, two articles were included in this systematic review.

The PRISMA-P diagram shown in Figure 1 summarizes the article selection steps and the characteristics of the excluded studies, where “DNMR” corresponds to “does not meet requirements”.

### 3.2. Characteristics of the Studies

The two studies included in the analysis, whose main characteristics are gathered in Appendix B, evaluated the relationship between the use of BZD and the development of AD [28,29]. The presence of insomnia in participants was assessed in both studies. The information regarding the diagnosis of insomnia in the study by Nafti et al. [29] was obtained by contacting the author. The two studies are characterized as cohorts, with the study by Lee et al. [28] being retrospective, and the study by Nafti et al. [29] prospective.

The duration of follow-up was 10.97 years in the study by Lee et al. [28], while in the study by Nafti et al. [29] the mean duration was 5.4 years (3.4–11.4 years). The two studies included a total of 273,451 participants, with 268,170 participants in the study by Lee et al. [28] and 5281 study participants by Nafti et al. [29].

The total number of participants with insomnia in the study by Nafti et al. [29] was 254 participants, with the control group consisting of 186 participants and the group that underwent treatment with BZDs consisting of 68 participants.

It should be noted that the study by Lee et al. [28] presents the number of participants diagnosed with insomnia according to “person-years”, totaling 2,844,200 person-years, with the control group consisting of 2,839,761 person-years and the group undergoing treatment with BZD consisting of 4439 person-years.

Participants’ age was equal to or greater than 50 years in the study by Lee et al. [28] and 65 or over in the study by Nafti et al. [29].

BZD treatment was evaluated by consulting existing data from the National Health Insurance Service (NHIS) in the study by Lee et al. [28]. In the study by Nafti et al. [29], treatment with BZDs was assessed using one of two ways: interview or questionnaire completed by the participant.

The study by Lee et al. [28] specifies the dose of BZD used, which was stratified according to the “Defined Daily Dose” (DDD) (average daily maintenance dose of a given active substance in its main therapeutic indication, in adults). Treatment with BZD was considered when DDD was equal to or greater than 30. Considering the possibility of reverse causality, participants were considered not to have been exposed to BZD up to 5 years after prescription > 30 DDD, only being considered exposed after this time.

The classification of BZD according to its half-life is detailed only in the study by Lee et al. [28], having verified the use of short-half-life BZD in 2221 person-years, intermediate-half-life in 1424 person-years, and long-half-life in 794 person-years.

Treatment with BZD lasted longer than one month in both studies. In the study by Lee et al. [28], it was considered that exposure to BZD occurred until the study was completed, while in the study by Nafti et al. [29] the end of exposure to BZD is undetermined since the treatment with BZD was evaluated through an interview or questionnaire, not being subject to further evaluation.

The diagnostic criteria for insomnia established by Lee et al. [28] were based on ICD-10. The criteria used in the study by Nafti et al. [29] were based on clinical history; the following parameters were questioned: “difficulty falling asleep”, “waking up earlier than desired”, “tendency to sleep all day”, and “feeling constantly tired”, with 104 participants presenting only initial insomnia, 75 participants only terminal insomnia, and 75 participants initial and terminal insomnia together.

The diagnostic criteria for AD established by Lee et al. [28] were based on ICD-10, and Nafti et al. [29] was based on the AD diagnostic criteria of the National Institute of Neurological and Communicative Diseases and Stroke/Alzheimer’s Disease and Related Disorders Association (NINCDS-ADRDA).

### 3.3. Quality

The quality of the analyzed studies is described in Table 1, where “Y” corresponds to “yes”, “N” corresponds to “no”, and “U” corresponds to “uncertain”. Two studies were included based on the described parameters.

### 3.4. Risk of Bias

The assessment of the risk of bias of the analyzed studies is detailed in Figure 2, carried out using the robvis (Risk Of Bias VISualization) tool [26].

The study by Nafti et al. [29] presents some risks and limitations, especially when compared to the study by Lee et al. [28], predominantly associated with the intervention and the potential limitation in the data to be evaluated.

### 3.5. Primary Output

#### Alzheimer’s Disease (AD)

The development of AD was the outcome evaluated in both studies.

Lee et al. [28] evaluated the presence of AD after treatment with BZD in patients with insomnia, finding a statistically significant positive association (*p* < 0.05) between these two variables (BZD and AD). Treatment was equal to or greater than 30 DDD, with a mean follow-up time of 10.97 years. The risk of developing AD was significantly higher when using hypnotics, regardless of the class of hypnotics used (HR: 1.79).

Regarding the specific treatment with BZD, three models were used to calculate the HR, the first referring to the HR without adjustment for comorbidities, the second adjusted for sex, DM, hypertension, hyperlipidemia, cerebrovascular disease, and individual health insurance value, and the third adjusted for anxiety, depression, psychotic disorders, and the comorbidities mentioned in the second model. In the group that underwent treatment with short-half-life BZD, HR values were obtained: 2.04 (model 1), HR: 2.05 (model 2), and HR: 2.00 (model 3). In the group that underwent treatment with BZD of intermediate half-life, HR values were obtained: 2.25 (model 1), HR: 2.18 (model 2), and HR: 2.01 (model 3). In the group that underwent treatment with long-half-life BZD, HR values were obtained: 2.06 (model 1), HR: 1.98 (model 2) and HR: 1.82 (model 3). Thus, there was an increased risk of developing AD when treating with BZD with an intermediate half-life (HR: 2.01) compared to treatment with BZD with a short half-life (HR: 2.00) and BZD with a long half-life (HR: 1.82), although the risk is significant regardless of the half-life. The HR values presented were adjusted for gender, and the following comorbidities: DM, hypertension, hyperlipidemia, cerebrovascular diseases, anxiety, and depression and psychiatric disorder, as well as the amount of individual health insurance.

Nafti et al. [29] evaluated the presence of AD after treatment with BZD in patients with insomnia, not finding an association between these two variables (BZD and AD). The treatment lasted more than one month, with an average follow-up time of 5.4 years. No association was demonstrated between AD incidence and BZD treatment. However, the study suggests that the effects of BZDs seem to exacerbate the clinical expression of early-stage dementia. It is highlighted that Nafti et al. [29] described a positive, statistically significant (*p* < 0.05) association between the development of “non-dementia” cognitive impairment and treatment with BZD in the total population evaluated, which includes participants with and without a diagnosis of insomnia. Three models were used to calculate the HR, the first adjusted for age (HR: 1.36), the second adjusted for gender and education level (HR: 1.38), and the third adjusted for smoking habits and alcohol, physical activity, depression, acute myocardial infarction, hypertension, stroke, DM, and treatment with non-steroidal anti-inflammatory drugs (HR: 1.32). However, no association was demonstrated between the development of AD and treatment with BZD in the total population evaluated, which includes participants with and without a diagnosis of insomnia. The three aforementioned models were used to calculate the HR, obtaining the following results: model 1 (HR: 0.89), model 2 (HR: 0.85), and model 3 (HR: 0.84).

### 3.6. Sample Characteristics

#### 3.6.1. Age

In the study by Nafti et al. [29], participants who underwent treatment with BZD were slightly older, although very similar (75.1 ± 6.7 years vs. 74.1 ± 6.4 years, *p* < 0.001), with an increase in the use of BZD parallel to the increase in age, with this being the only study to present these data.

#### 3.6.2. Sex

In the study by Lee et al. [28], the female sample that underwent treatment with BZD was superior compared to the control group (56.21% vs. 43.70%, *p* < 0.01), with the study by Nafti et al. [29] presenting a similar trend (72.8% vs. 59.8%, *p* < 0.001), so that treatment with BZD was predominant in females in both studies.

#### 3.6.3. Housing Conditions

In the study by Nafti et al. [29], it was found that treatment with BZD showed a slight predominance in institutionalized patients, with 12% (*p* < 0.001) of participants who underwent treatment with BZD being institutionalized compared to the control group where only 3% (*p* < 0.001) of the participants were institutionalized, with this being the only study to present these data.

#### 3.6.4. Education Level

Treatment with BZD predominated in participants with a lower level of education (9.6 ± 4.1 years of schooling, *p* < 0.001) compared to the control group (10.8 ± 3.7 years of schooling, *p* < 0.001) in the study by Nafti et al. [29], and this was the only study to present these data.

#### 3.6.5. Socioeconomic Status

Treatment with BZD predominated in participants with lower socioeconomic status in the study by Lee et al [28]. Despite being the only study to present these data, it does not present the corresponding percentage value.

#### 3.6.6. Comorbidities

Participants who underwent treatment with BZD had a greater number of comorbidities, compared to the control group, in both studies.

In the study by Nafti et al. [29], the following comorbidities were evaluated: depression (21% vs. 7%, *p* < 0.001) and vascular diseases, namely, acute myocardial infarction (33% vs. 24%, *p* < 0.001), hypertension (40% vs. 35%, *p* = 0.03), and stroke (12% vs. 7%, *p* < 0.001).

In the study by Lee et al. [28], the evaluated comorbidities, with a statistically significant value, were the following: DM (26.63% vs. 18.62%, *p* < 0.01), AHT (61.83% vs. 43.25%, *p* < 0.01), hyperlipidemia (35.38% vs. 18.76%, *p* < 0.01), cerebrovascular disease (14.28% vs. 7.46%, *p* < 0.01), anxiety (25.55% vs. 3.49%, *p* < 0.01), and depression (10.07% vs. 1.84%, *p* < 0.01).

#### 3.6.7. Medication

In the study by Nafti et al. [29], participants who underwent treatment with BZD had higher levels of treatment with non-steroidal anti-inflammatory drugs compared to the control group (59% vs. 53%, *p* < 0.01), and this was the only study to present these data.

## 4. Discussion

In the present systematic review, we sought to deepen the current knowledge about the association of long-term insomnia treatment with BZD and AD. Therefore, a systematic review was carried out in order to obtain a comprehensive view of the possible deleterious effects resulting from long-term treatment with BZD and, more specifically, the risk of developing AD when treated with this pharmacological class, in the presence of insomnia. Only two cohort studies were in agreement with the eligibility criteria and were included in the narrative synthesis of results. The retrospective study by Lee et al. [28] found a significant risk of developing AD during treatment with long-term use of BZD, regardless of the half-life of BZD, while in the prospective study by Nafti et al. [29] the incidence of AD did not correlate with BZD treatment. It should be noted, however, that Nafti et al. [29] demonstrated a positive association between the development of cognitive impairment and BZD treatment.

The analysis of the population studied in both studies made it possible to verify that treatment with BZD has a higher prevalence in females. On the other hand, those who underwent treatment with BZD had a greater number of comorbidities, including depression, cardiovascular disease, and hypertension. It should be noted that the female gender and the comorbidities presented are risk factors identified for the development of AD [6,30,31]. Of the results evaluated, only in the study by Nafti et al. [29] was it possible to verify that treatment with BZD increased with increasing age, with a predominance in institutionalized participants and those with a lower level of education. Treatment with nonsteroidal anti-inflammatory drugs was superior in those undergoing treatment with BZD. Considering the presence of a greater number of comorbidities that, by themselves, constitute risk factors for the development of AD, treatment with BZD represents an increased risk in this population. Of the results evaluated, only in the study by Lee et al. [28] was it found that the sample that underwent treatment with BZD had a lower socioeconomic status. A lower socioeconomic status is associated with a lower level of education and, as this constitutes a modifiable risk factor for AD, it is possible that this characteristic interferes with the relationship with the development of AD, as demonstrated by Gauthier et al. [31].

Difficulties are notorious regarding the quantity and quality of existing studies to date. Despite the high prevalence of BZD treatment, studies on this pharmacological class are scarce. The samples of the analyzed studies (Lee et al. [28]; Nafti et al. [29]) showed a significantly different number of participants with insomnia (7862 vs. 254), not allowing a rigorous comparison of the results. It should be noted that given the presentation of data from participants with insomnia in the study by Lee et al. [28] according to “person-years”, there is also a greater difficulty when comparing the samples of the two studies. The follow-up time of each of the studies was also different (5.4 years vs. 10.97 years), which may have biased the results presented given that, in the study that did not find a relationship between BZD and AD [29], the average follow-up time was significantly lower (approximately half) compared to the study that verified the association between BZD and AD [28].

On the other hand, the diagnostic criteria used with regard to insomnia and AD were different in both studies, presenting different sensitivities and specificities, thus reducing the effectiveness of the joint analysis of the results. The study by Nafti et al. [29] only used clinical criteria when diagnosing insomnia, and it was the study that did not find significant results regarding the association under analysis, contrary to the study by Lee et al. [28], who performed the diagnosis of insomnia according to the ICD-10 criteria. The use of distinctive diagnostic criteria for insomnia and AD can impact the interpretation and reliability of the results. This added heterogeneity can challenge comparisons as questions may arise related to diagnostic reliability, as ICD-10 criteria may be more able to standardize diagnoses across larger datasets, while clinical criteria may capture subtler cases, introducing possible bias. Future research should adopt standardized diagnostic criteria to enhance comparability and reduce variability.

It should also be noted that the analyzed studies were observational (cohorts), not obeying the methodological rigor of randomized clinical trials, so there are limitations associated with them, such as the loss of follow-up. Unfortunately, this type of study is limited in its ability to provide us with more definitive answers. Several factors may constrain its reliability: the inability to fully control for confounding variables such as physical activity, comorbidities, or socioeconomic status, which can influence outcomes and may not be adequately addressed; the potential for reverse causation, as observational studies often struggle to establish the temporal sequence between exposure and outcome; and the inherent inability to confirm causation, as these studies can only identify associations and lack the randomization necessary to draw causal conclusions [32,33,34]. In this way, the lack of studies with an intervention that allows the confirmation of the hypothesis that is being posed is highlighted.

The study by Lee et al. [28] considered a lag period of 5 years between the beginning of treatment with BZD and the moment when the sample was considered exposed to the treatment. However, it recognizes as a limitation the possibility of reverse causality, given the possibility that prodromal symptoms of AD, such as insomnia, may appear in a period prior to the onset of symptomatic dementia of more than ten years. The possibility of reverse causality was also mentioned in the study by Nafti et al. [29].

Taking into account that in the study by Lee et al. [28] treatment with BZD was evaluated by consulting existing prescription data in the NHIS, there may have been a significant difference between the prescription of BZD and its administration.

Regarding the quality of the articles, in the study by Lee et al. [28] only one of the eleven questions evaluated obtained a result compatible with a potential risk of bias, due to the differences between the group that underwent treatment with BZD and the control group. In the study by Nafti et al. [29], four of the eleven questions evaluated obtained results that potentiate the risk of bias. The group that underwent treatment with BZD and the control group had characteristics that differed from each other, exposure to BZD was assessed through an interview or questionnaire completed by the participant, which may not have reflected exposure to treatment with BZD, the follow-up time average up was 5.4 years, which may not have been sufficient for the development of AD, and a high percentage was lost to follow-up in relation to the initial sample.

Regarding the risk of bias, both studies have a moderate risk of confounding bias. The study by Lee et al. [28] did not include factors related to lifestyles, such as smoking and drinking habits. In the study by Nafti et al. [29], the presence of anxiety, traumatic brain injury, and exposure to anticholinergics or other sleep disorders was not evaluated. It is also considered that the factors evaluated, namely, hypertension, DM, and treatment with non-steroidal anti-inflammatory drugs, may have been subject to modification during the follow-up period, which may partly support the results presented.

The study by Lee et al. [28], being retrospective, presents a moderate risk of bias in the evaluation of the results, given the a priori knowledge of the participants who underwent treatment with BZD. The study by Nafti et al. [29] presents a serious risk of bias in the classification of the studied intervention. The authors of the study under analysis mention that the treatment with BZD was evaluated through an interview or questionnaire completed by the participant, and this information was not subject to further validation, as it lacked information on the pattern of BZD use as well as the dose and intermittent exposure or exposure to a multiple number of BZDs, so the results may have been underestimated.

The study by Nafti et al. [29] refers to the withdrawal of a high percentage (27.2%) of the sample at the beginning of the study. These participants had a higher prevalence of BZD treatment, which may have underestimated the results, resulting in a critical risk of bias due to lack of data.

Despite the identified limitations associated with the study by Nafti et al. [29], we chose to include it in the present review for several reasons: (i) the relevance of these data in view of the scarcity verified in the scientific literature; (ii) the type of study is particularly important for the topic in question. Given the above, it is considered that the study by Lee et al. [28] presents a larger population, a lower risk of bias, and higher quality, so the results obtained in this study need to be further valued, given the higher level of confidence attributed to it, compared to the study by Nafti et al. [29].

As for the existing literature, the data are equally contradictory in studies that dealt with the same topic, but that did not meet the inclusion criteria: Imfeld et al. [35] did not identify an association between long-term use of BZD and the development of AD. They also described that the possible association demonstrated in the existing literature could be related to the use of BZD in the early stages of dementia, which would justify the results found. Gray et al. [36] did not identify a significant association between BZD use and the development of AD regardless of exposure level; however, Jeong et al. [37] identified an increased risk of developing AD when using long-half-life BZD, highlighting that the risk of dementia is also related to the increased serum concentration of long-half-life BZD in the elderly.

Changes in the sleep–wake cycle and circadian rhythm occur in an early stage of AD in about 40% of patients [38]. It is of particular relevance to emphasize that insomnia can present itself as a prodrome of AD, and can also be a manifestation of DA [38,39,40]. On the other hand, insomnia is associated with the development of cognitive impairment, and the relationship between insufficient sleep quality and cortical atrophy has been described, whereby there is a bidirectional relationship between insomnia and brain disorders [11]. An association between slow-wave sleep disruption and increased beta-amyloid levels was demonstrated, establishing sleep deprivation as a risk factor for dementia [39]. Pase et al. [41] demonstrated that since cholinergic neurons are activated during REM sleep and, since the loss of cholinergic functioning is a characteristic of AD, the alteration of REM sleep constitutes an argument in favor of the development of dementia, verifying a similarly bidirectional relationship between insomnia and, more specifically, AD [39,41,42]. This dichotomy is manifested by the difficult distinction between the etiology and manifestations of the disease.

This systematic review emerges in the face of the lack of more reliable literature regarding the association between long-term treatment of insomnia with BZD and the development of AD. Considering the results presented, given the scarcity of findings, it does not seem possible to establish a robust and consistent association between the two variables studied, so the risk of developing AD when using BZD in the presence of insomnia remains uncertain. However, results continue to support a possible association between AD and BZD use in insomnia. The study with the longest follow-up shows a significant risk of developing AD, which is why it constitutes the necessary premise to continue the study of this issue.

## 5. Conclusions

Long-term treatment with BZD is not recommended given the lack of evidence and the possibility of adverse effects; however, it was possible to verify that it has a high prevalence, so the study of its implications is of special importance. AD is the main cause of dementia, requiring curative treatment, which is why it remains essential to elucidate possible modifiable risk factors.

In view of the small number of studies carried out in this area and given the relevance of this theme, the need for future research is urgent. It is of special importance to carry out new studies, giving primacy to the necessary rigor and accuracy. In this sense, there is a need for prospective longitudinal studies with larger samples, with a longer and similar follow-up time, as well as a thorough control of potential confounding factors. It is also important to emphasize the need to standardize the diagnostic criteria for insomnia and AD. Treatment with BZD also lacks a more accurate and reliable evaluation. Considering that insomnia can be a prodromal symptom of AD, the possibility of reverse causality should be considered when designing future studies.

## Figures and Tables

**Figure 1 neurosci-06-00011-f001:**
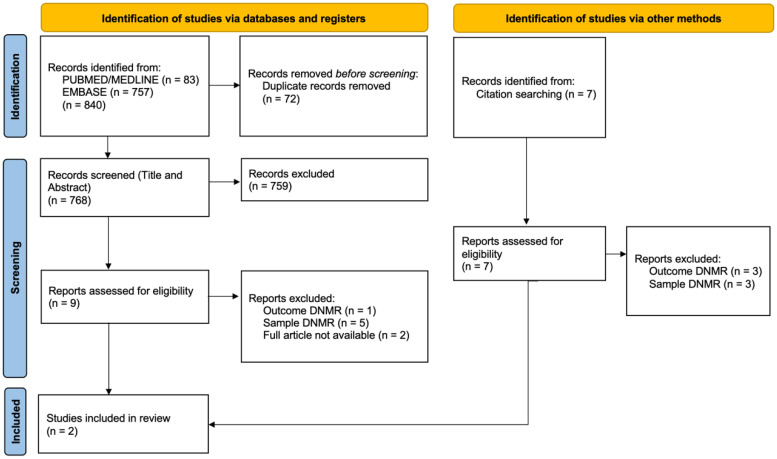
PRISMA-P 2020 diagram for the process of the selection of studies.

**Figure 2 neurosci-06-00011-f002:**
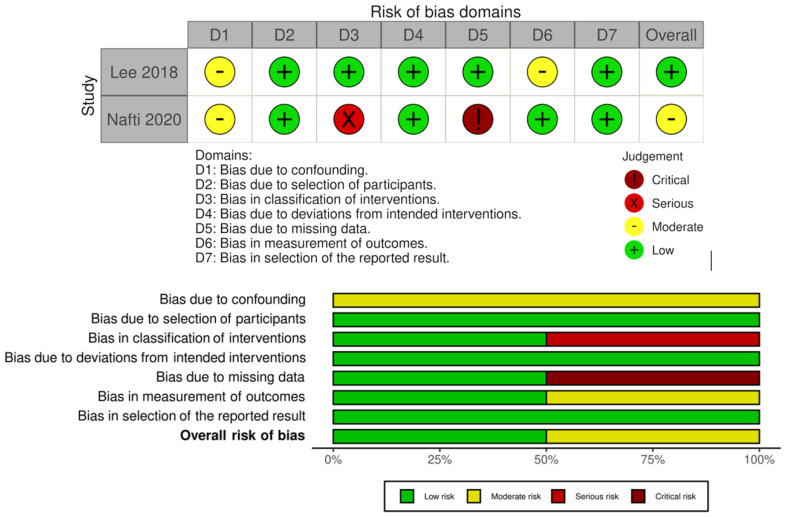
Evaluation of the different risk factors for bias and bias risk in percentual values from the two studies included in this systematic review, Lee et al. [28] and Nafti et al. [29].

**Table 1 neurosci-06-00011-t001:** Quality parameter evaluation from the articles included (questionary of quality evaluation of cohort studies—JBI; questions included as Appendix C).

Author, Year	Questions											Decision
	1	2	3	4	5	6	7	8	9	10	11	
Lee, 2018 [28]	N	Y	Y	Y	Y	Y	Y	Y	Y	Y	Y	Included
Nafti, 2020 [29]	N	Y	N	Y	Y	Y	Y	U	Y	N	Y	Included

## Data Availability

Data will be made available on request.

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
