# Peer review of "Long-Term Insomnia Treatment with Benzodiazepines and Alzheimer’s Disease: A Systematic Review"

_neurosci, 2025, doi:10.3390/neurosci6010011_

Round 1
Reviewer 1 Report
Comments and Suggestions for Authors
The article by Filipa Sofia Trigo and colleagues provides a systematic review of long-term insomnia treatment with benzodiazepines and Alzheimer's disease. In this article, the authors have mainly focused on the role of benzodiazepines and their association with the development of Alzheimer's disease. The authors have used PRISMA-P 2020 guidelines, MEDLINE, and Embase database. This is a well-written article that provides the details of benzodiazepines and AD, and results suggest that benzodiazepines long-term treatment is not recommended, given the lack of evidence and the possibility of adverse effects. The study suggests that further research is essential before concluding the benzodiazepine role in Alzheimer's disease. However, the overall quality of the article should be improved, and changes should be made.
The authors should increase the font sizes in Figure 1 for better representation.
In this article, the authors mention the limitations of specific studies; however, separate limitations should be mentioned. This will help to get the current advancements and limitations of this field.
In the result section, it was mentioned that (3.3. Quality- The quality of the analyzed studies is described in Table 1, where “Y” corresponds to “yes”, “N” corresponds to “no” and “U ” corresponds to “uncertain.”) In Table 1, the authors should provide a detailed list of 1-11 questions. Also, the Table description should provide “S” and “I” details.
Figure 2 needs more description in the Figure legends.
Here, the authors have already added the Table for two specific references that were included in this study. However, the current study discusses different article findings, and the authors should consider adding these findings to the new table.
Author Response
Authors Responses
Reviewer 1
The article by Filipa Sofia Trigo and colleagues provides a systematic review of long-term insomnia treatment with benzodiazepines and Alzheimer's disease. In this article, the authors have mainly focused on the role of benzodiazepines and their association with the development of Alzheimer's disease. The authors have used PRISMA-P 2020 guidelines, MEDLINE, and Embase database. This is a well-written article that provides the details of benzodiazepines and AD, and results suggest that benzodiazepines long-term treatment is not recommended, given the lack of evidence and the possibility of adverse effects. The study suggests that further research is essential before concluding the benzodiazepine role in Alzheimer's disease. However, the overall quality of the article should be improved, and changes should be made.
Comment 1:The authors should increase the font sizes in Figure 1 for better representation.
Response 1: We thank the reviewer for the suggestion, which we agree with. We've made a new PRISMA flow diagram with a larger font size. Please see revised manuscript as Fig.1.
Comment 2: In this article, the authors mention the limitations of specific studies; however, separate limitations should be mentioned. This will help to get the current advancements and limitations of this field.
Response 2: We thank the reviewer for this comment. We believe that we also mentioned this topic in the future studies to be done. As noted, cohort studies do have several limitations, and we also increased this subject and this overall topic in the discussion - Please see revised manuscript.
Comment 3: In the result section, it was mentioned that (3.3. Quality- The quality of the analyzed studies is described in Table 1, where “Y” corresponds to “yes”, “N” corresponds to “no” and “U ” corresponds to “uncertain.”) In Table 1, the authors should provide a detailed list of 1-11 questions. Also, the Table description should provide “S” and “I” details.
Response 3: Thank you for your pertinent comments. In fact, the letters S and N have not been correctly translated in the table in question, and should have been replaced by the letters Y and N respectively (for Sim=Yes and Não=No). We have therefore corrected the table in the revised manuscript. As for the questions, they are the same as those on the JBI form, without any changes - for this reason we haven't included them in the main manuscript so as not to increase the size of the text excessively. At the reviewer's request, we have opted to add them to the appendices of the new manuscript.
Comment 4: Figure 2 needs more description in the Figure legends.
Response 4: We thank the reviewer for the suggestion. In order to improve and complete the information provided, we have added more specific data for each item included. We hope that this change will allow you to better understand the analysis of the articles. Please see the revised manuscript.
Comment 5: Here, the authors have already added the Table for two specific references that were included in this study. However, the current study discusses different article findings, and the authors should consider adding these findings to the new table.
Response 5: Unfortunately, we can't find on the website or in the manuscript with notes to correct which part of the main text the reviewer is referring to. If the reviewer or editor feels it is appropriate to respond to this point, we would ask for more information and the location in the text where the reviewer asks for improvements to be made. We will be happy to respond and/or improve the text.
Reviewer 2 Report
Comments and Suggestions for Authors
The paper provides a well-organized and comprehensive review of the association between benzodiazepine (BZD) use and AD, with an emphasis on long-term BZD use for treating insomnia. Below are suggestions for improving clarity, structure, and comprehensiveness of the article:
In some places, the terms "insomnia" and "sleep disorders" are used interchangeably. It might be helpful to distinguish more clearly between general sleep disturbances and insomnia to avoid confusion. Insomnia has specific diagnostic criteria and should be treated as a distinct entity.
Consider adding a brief definition of "sleep disorders" at the first mention, so the reader understands the broader category under which insomnia falls.
The flow of information could be improved by reorganizing some sections. For example, the section on the Pathophysiology of AD could be placed earlier in the introduction, before the discussion on insomnia and BZD use. This will help set the context for understanding why this relationship is of particular interest.
In the section on the Strategy for Data Synthesis, there is a discussion of hazard ratios (HR) and odds ratios, but it might be beneficial to briefly explain the distinction between these two metrics. This could aid readers who may not be familiar with statistical methodologies.
The introduction mentions the prevalence of AD as affecting "around 25-50 million people" this is a very wide range. It might be helpful to specify the source of this statistic or revise it to reflect more recent and precise data.
Causal relationship between insomnia, BZD use, and AD: While the potential link between insomnia, BZD use, and AD is clearly outlined, the introduction might benefit from further discussion on how BZD might contribute to AD pathology (e.g., through cognitive impairment or neurodegeneration).
Clarify Study Design Limitations:
Although the authors mention the observational nature of the studies (Lee et al. [24] and Nafti et al. [25]), a more detailed discussion about the specific limitations of observational studies would be beneficial. For example, more emphasis could be placed on the lack of control over confounding factors inherent in cohort studies, and how this affects the ability to draw causal conclusions.
It would strengthen the review to include more on the potential mechanisms by which BZD treatment could increase the risk of AD. For example, how does BZD interact with neurotransmitter systems (e.g., GABA receptors) and might this play a role in the development of AD? Explaining the possible biological pathways would provide readers with a better understanding of the connection.
The authors briefly mention contradictory findings in the literature (e.g., Imfeld et al. [28], Gray et al. [29], and Jeong et al. [30]), but the discussion around these studies could be expanded. Specifically, it would be valuable to interpret why these studies found different results (e.g., differences in study design, population, dose of BZD used, or diagnostic criteria for AD).
More detail could be provided about specific confounding factors that may have affected the studies. For instance, how did the presence of comorbidities like depression, hypertension, or cardiovascular disease influence the results? Could any other factors, such as genetics or environmental factors, be influencing the relationship between BZD and AD?
Since the studies used different diagnostic criteria for insomnia and AD (ICD-10 vs clinical criteria), the authors could elaborate more on how this variability impacts the overall interpretation of the results. They could discuss the possible consequences of using non-standardized diagnostic criteria on study reliability and validity.
Recommendations for Future Research:
While the authors call for future research, they could provide more specific recommendations. For example, they could discuss potential study designs (e.g., randomized controlled trials, large cohort studies) or suggest more precise measures for insomnia and AD diagnoses. Additionally, it would be helpful to discuss possible inclusion criteria for studies in future systematic reviews to help reduce heterogeneity and improve the generalizability of findings.
Author Response
Authors Responses
Reviewer 2
The paper provides a well-organized and comprehensive review of the association between benzodiazepine (BZD) use and AD, with an emphasis on long-term BZD use for treating insomnia. Below are suggestions for improving clarity, structure, and comprehensiveness of the article:
Comment 1: In some places, the terms "insomnia" and "sleep disorders" are used interchangeably. It might be helpful to distinguish more clearly between general sleep disturbances and insomnia to avoid confusion. Insomnia has specific diagnostic criteria and should be treated as a distinct entity. Consider adding a brief definition of "sleep disorders" at the first mention, so the reader understands the broader category under which insomnia falls.
Response 1: We thank the reviewer for the pertinent comment. We agree with the reviewer and went on to re-evaluate all the mentions of “sleep disorders” throughout the text, also re-evaluating in what context they appear in the articles cited. We modified the text or replaced the term in order to increase scientific rigor. In the revised manuscript, only 2 mentions of “sleep disorders” remained, which we now believe will not cause any misinterpretation. Please see the revised manuscript (abstract and discussion).
Comment 2: The flow of information could be improved by reorganizing some sections. For example, the section on the Pathophysiology of AD could be placed earlier in the introduction, before the discussion on insomnia and BZD use. This will help set the context for understanding why this relationship is of particular interest.
Response 2: We thank the reviewer for this comment. We reviewed the introduction, and we think that it follows the reviewer´s idea. However, if we did not understand the point, we will be happy to reformulate it again.
Comment 3: In the section on the Strategy for Data Synthesis, there is a discussion of hazard ratios (HR) and odds ratios, but it might be beneficial to briefly explain the distinction between these two metrics. This could aid readers who may not be familiar with statistical methodologies.
Response 3: We agree with the reviewer and have added the requested information - please see section 2.6. Strategy for data synthesis of the revised manuscript.
Comment 4: The introduction mentions the prevalence of AD as affecting "around 25-50 million people" this is a very wide range. It might be helpful to specify the source of this statistic or revise it to reflect more recent and precise data.
Response 4: We thank the reviewer for the pertinent comment. We agree with the reviewer and have updated the requested information in the beginning of the introduction - please see the revised manuscript.
Comment 5: Causal relationship between insomnia, BZD use, and AD: While the potential link between insomnia, BZD use, and AD is clearly outlined, the introduction might benefit from further discussion on how BZD might contribute to AD pathology (e.g., through cognitive impairment or neurodegeneration).
Response 5: We thank the reviewer for the pertinent comment. We agree with the reviewer and have added the requested information in the introduction - please see the revised manuscript.
Comments 6:
Clarify Study Design Limitations:
Although the authors mention the observational nature of the studies (Lee et al. [24] and Nafti et al. [25]), a more detailed discussion about the specific limitations of observational studies would be beneficial. For example, more emphasis could be placed on the lack of control over confounding factors inherent in cohort studies, and how this affects the ability to draw causal conclusions.
Response 6.1: We agree with the reviewer and have added the requested information in the discussion - please see the revised manuscript.
It would strengthen the review to include more on the potential mechanisms by which BZD treatment could increase the risk of AD. For example, how does BZD interact with neurotransmitter systems (e.g., GABA receptors) and might this play a role in the development of AD? Explaining the possible biological pathways would provide readers with a better understanding of the connection.
Response 6.2: We agree with the reviewer and have addressed this topic in the introduction - please see the revised manuscript. However, we have chosen not to go into too much detail so as not to lengthen this article, which is already long for the journal.
The authors briefly mention contradictory findings in the literature (e.g., Imfeld et al. [28], Gray et al. [29], and Jeong et al. [30]), but the discussion around these studies could be expanded. Specifically, it would be valuable to interpret why these studies found different results (e.g., differences in study design, population, dose of BZD used, or diagnostic criteria for AD).
Response 6.3: Thank you for your suggestion. We understand the reviewer's point of view - a broader discussion could be fruitful, but the approach written in the main text already takes up a significant portion of the main text and we think that extending it could detract from the reading of the article. However, if the reviewer thinks that discussing these 3 articles not included in the review is of major relevance, we would be happy to do so.
More detail could be provided about specific confounding factors that may have affected the studies. For instance, how did the presence of comorbidities like depression, hypertension, or cardiovascular disease influence the results? Could any other factors, such as genetics or environmental factors, be influencing the relationship between BZD and AD?
Response 6.4: Thank you for your suggestion. We believe that this topic has been covered in the main text, which we'll post below:
“Regarding the risk of bias, both studies have a moderate risk of confounding bias. The study by Lee et al. [28] did not include factors related to lifestyles, such as smoking and drinking habits. In the study by Nafti et al. [29] the presence of anxiety, traumatic brain injury, and exposure to anticholinergics or other sleep disorders was not evaluated. It is also considered that the factors evaluated, namely, hypertension, DM, and treatment with non-steroidal anti-inflammatory drugs, may have been subject to modification during the follow-up period, which may partly support the results presented.”
If the reviewer thinks that further discussion in needed, we would be happy to do so.
Since the studies used different diagnostic criteria for insomnia and AD (ICD-10 vs clinical criteria), the authors could elaborate more on how this variability impacts the overall interpretation of the results. They could discuss the possible consequences of using non-standardized diagnostic criteria on study reliability and validity. While the authors call for future research, they could provide more specific recommendations. For example, they could discuss potential study designs (e.g., randomized controlled trials, large cohort studies) or suggest more precise measures for insomnia and AD diagnoses. Additionally, it would be helpful to discuss possible inclusion criteria for studies in future systematic reviews to help reduce heterogeneity and improve the generalizability of findings.
Response 7: We thank the reviewer for the pertinent comment. We tried to address this topics and reformulated parts of the discussion - please see the revised manuscript.